# Near-Optimal Thompson Sampling-based Algorithms for Differentially Private Stochastic Bandits

**Bingshan Hu**[1,2]                                **Nidhi Hegde**[1,2]

[1]Department of Computing Science, University of Alberta, Edmonton, Alberta, Canada
[2]Amii (Alberta Machine Intelligence Institute)

## Abstract

We address differentially private stochastic bandits. We present two (near)-optimal Thompson Sampling-based learning algorithms: DP-TS and Lazy-DP-TS. The core idea in achieving optimality is the principle of optimism in the face of uncertainty. We reshape the posterior distribution in an optimistic way as compared to the non-private Thompson Sampling. Our DP-TS achieves a $\sum_{j \in \mathcal{A}: \Delta_j > 0} O\left( \frac{\log(T)}{\min\{\epsilon, \Delta_j\}} \log\left( \frac{\log(T)}{\epsilon \cdot \Delta_j} \right) \right)$ regret bound, where $\mathcal{A}$ is the arm set, $\Delta_j$ is the sub-optimality gap of a sub-optimal arm $j$, and $\epsilon$ is the privacy parameter. Our Lazy-DP-TS gets rid of the extra log factor by using the idea of dropping observations. The regret of Lazy-DP-TS is $\sum_{j \in \mathcal{A}: \Delta_j > 0} O\left( \frac{\log(T)}{\min\{\epsilon, \Delta_j\}} \right)$, which matches the regret lower bound. Additionally, we conduct experiments to compare the empirical performance of our proposed algorithms with the existing optimal algorithms for differentially private stochastic bandits.

## 1 INTRODUCTION

We consider the setting of differentially private stochastic multi-armed bandits[Mishra and Thakurta, 2015, Tossou and Dimitrakakis, 2016, Shariff and Sheffet, 2018, Sajed and Sheffet, 2019, Hu et al., 2021]. In the classical stochastic multi-armed bandit problem, we have a fixed and finite set of $K$ arms and a stochastic environment. In each round $t = 1, 2, \ldots, T$, the environment generates a random reward $X_j(t)$ for arm $j$ which is revealed and collected if arm $j$ is pulled in that round. For each arm $j$, the rewards $X_j(t) \in [0, 1]$ are i.i.d. over time according to a fixed but unknown probability distribution with mean $\mu_j$. The goal of

the learning algorithm is to pull arms sequentially to maximize the accumulated reward. The performance metric has traditionally been pseudo-*regret* [Bubeck and Cesa-Bianchi, 2012] which is a measure of the difference of the expected accumulated rewards compared to a given benchmark.

In the classical setting, the learning algorithm uses the true revealed rewards from previous rounds to make decisions on arms in future rounds. However, in many settings rewards may be private information that should be protected. For instance, consider an online search advertisement system where the objective is to display relevant ads for web queries. In such a setting the system would display a few advertisements to the user. When the user clicks on an ad, a reward is collected by the system, which in accumulation would allow the system and any external observers to learn user preferences. Rewards that represent user preferences are private information and may further allow inference on the user's other private characteristics.

Motivated by such applications, previous work [Mishra and Thakurta, 2015, Tossou and Dimitrakakis, 2016, Sajed and Sheffet, 2019, Hu et al., 2021] have studied the design of bandit learning algorithms with differential privacy[Dwork et al., 2014] for keeping reward information private. Differential privacy has been used as a framework because it provides robust privacy guarantees and a controlled tradeoff with regret guarantees in the case of bandit learning.

Two of the most common algorithms in the stochastic bandit setting are Upper Confidence Bound (UCB) sampling [Auer et al., 2002] and Thompson Sampling [Agrawal and Goyal, 2017]. Mishra and Thakurta [2015] present the first differentially private versions of these two algorithms and provide regret bounds for their algorithms. This was followed by differentially private algorithms in the contextual linear bandit setting [Shariff and Sheffet, 2018] and optimal differentially private algorithms based on Successive Elimination (DP-SE) [Sajed and Sheffet, 2019] and UCB (Anytime-Lazy-UCB) [Hu et al., 2021] in the stochastic bandit setting.

However, to the best of our knowledge, there is still no opti-

*Accepted for the 38th Conference on Uncertainty in Artificial Intelligence* (UAI 2022).

mal Thompson Sampling-based algorithm for differentially private stochastic bandits. Thompson Sampling-based algorithms exhibit better performance than UCB or SE-based methods, are applicable to a wider range of information models, and are more widely implemented in practical scenarios [Chapelle and Li, 2011, Gopalan et al., 2014]. Given their widespread implementation in practice, we provide two (near)-optimal Thompson Sampling-based algorithms for private stochastic bandits. The core concept in our algorithms still relies on the principle of optimism in the face of uncertainty. More specifically, we shift the posterior distribution of an arm to the right as compared to the posterior distribution in the non-private Thompson Sampling.

Our first algorithm, Differentially Private Thompson Sampling (DP-TS), can be viewed as a differentially private version of the standard Thompson Sampling for Bernoulli bandits [Agrawal and Goyal, 2017]: the learning algorithm makes a decision based on all observations obtained from the beginning and updates the statistics of the pulled arm at the end of each round. The regret bound of DP-TS is $\widetilde{O}\left(\frac{K \log(T)}{\min\{\epsilon, \Delta\}}\right)$, where $\widetilde{O}(\cdot)$ hides an extra $\log(\log(T)/(\epsilon\Delta))$ factor. Our second algorithm, Lazy Differentially Private Thompson Sampling (Lazy-DP-TS), drops observations during learning and updates the statistics of the pulled arm in a delayed manner. With these modifications we achieve the optimal $O\left(\frac{K \log(T)}{\min\{\Delta, \epsilon\}}\right)$ regret bound. Interestingly, as discussed in Section 3 and confirmed in Section 4 with numerical experiments, DP-TS may perform better than Lazy-DP-TS under some circumstances.

**Contribution.** We make the following key contributions. (1) We present (near)-optimal Thompson Sampling-based learning algorithms for differentially private stochastic bandits: DP-TS and Lazy-DP-TS; (2) The regret bound for DP-TS is $\sum_{j \in \mathcal{A}: \Delta_j > 0} O\left(\max\left\{\frac{\log(T)}{\Delta_j}, \frac{\log(T)}{\epsilon} \log\left(\frac{\log(T)}{\epsilon \cdot \Delta_j}\right)\right\}\right)$, which is optimal up to a $\log\log(T)$ factor (Theorem 2); (3) The regret bound for Lazy-DP-TS is $\sum_{j \in \mathcal{A}: \Delta_j > 0} O\left(\frac{\log(T)}{\min\{\epsilon, \Delta_j\}}\right)$, which is optimal (Theorem 4); (4) We show through numerical experiments performance improvement of our proposed learning algorithms as compared to the existing two optimal algorithms, DP-SE and Anytime-Lazy-UCB.

## 2 PROBLEM DEFINITION AND BACKGROUND

### 2.1 STOCHASTIC MULTI-ARMED BANDITS

We consider the stochastic multi-armed bandit setting, with a fixed set $\mathcal{A}$ of $K$ arms and a stochastic environment. At each round $t = 1, 2, \ldots, T$, the environment generates a reward vector $X_t := (X_1(t), X_2(t), \ldots, X_K(t))$ with each $X_j(t) \in \{0, 1\}$ independently drawn from a Bernoulli distri-

bution[1] with parameter $\mu_j \in (0, 1)$. The learning algorithm pulls an arm $J_t \in \mathcal{A}$ and at the end of round $t$, observes and obtains the reward of the pulled arm, $X_{J_t}(t)$. The goal of the algorithm to select an arm in each round such that the accumulated reward over $T$ rounds is maximized.

Without loss of generality, we assume that the optimal arm is unique and let arm 1 be the unique optimal arm, i.e., $\mu_1 > \mu_j$ for all $j \in \mathcal{A}\backslash\{1\}$. Let $\Delta_j := \mu_1 - \mu_j$ be the mean reward gap, which indicates the performance loss in a single round when a sub-optimal arm $j$ is pulled instead of the best arm 1. We use (pseudo)-regret $\mathcal{R}(T)$ to measure the performance, expressed as

$$
\begin{aligned}
\mathcal{R}(T) &= T \cdot \mu_1 - \sum_{j \in \mathcal{A}} \mathbb{E}\left[\sum_{t=1}^{T} \mathbf{1}\{J_t = j\}\right] \cdot \mu_j \\
&= \sum_{j \in \mathcal{A}: \Delta_j > 0} \mathbb{E}\left[\sum_{t=1}^{T} \mathbf{1}\{J_t = j\}\right] \cdot \Delta_j .
\end{aligned}
$$

### 2.2 DIFFERENTIAL PRIVACY

Differential privacy is a widely accepted framework of privacy and is based on the notion of plausible deniability: an adversary should learn nearly the same thing if one element in the dataset is changed or missing. In the context of bandits, a dataset is the stream of reward vectors drawn throughout the algorithm, and a change would refer to one reward vector in the stream. More formally, let $X_{1:t}$ be the sequence of reward vectors up to time $t$ and let $X'_{1:t}$ be a neighbouring sequence which differs in at most one reward vector, say, at any round $s, s \leq t$. The output of a bandit learning algorithm is the sequence of arm selections at each round. In this context, differential privacy is defined as follows, omitting the subscript $t$ for clarity.

**Definition 1.** *An online learning algorithm $\mathcal{M}$ is $\epsilon$-differentially private if, at every round $t = 1, \ldots T$, for any two neighbouring reward sequences $X$ and $X'$, and for any set $\mathcal{D}$ of decisions made, it holds that $\mathbb{P}\{\mathcal{M}(X) \in \mathcal{D}\} \leq e^{\epsilon} \cdot \mathbb{P}\{\mathcal{M}(X') \in \mathcal{D}\}$.*

**Remark.** Our definition of differential privacy follows the standard notion that was introduced in [Dwork et al., 2014]. It can also be interpreted as and is very related to the Max Divergence $D_\infty(Q, Q') := \max_{y \in \text{Support}(Q')} \ln \frac{\mathbb{P}(Q=y)}{\mathbb{P}(Q'=y)}$ between two probability distributions $Q$ and $Q'$. If we view $Q$ as the output distribution (the distribution of the sequentially pulled arms) when working over the true reward sequences $X$ and view $Q'$ as the output distribution when working over $X'$, an $\epsilon$-differentially private algorithm guarantees that the maximum divergence between $Q$ and $Q'$ is at most $\epsilon$, i.e., $\ln\left(\frac{\mathbb{P}\{Q=y\}}{\mathbb{P}\{Q'=y\}}\right) \leq \epsilon$ for all possible output $y$. Actually, the quantity $\ln\left(\frac{\mathbb{P}\{Q=y\}}{\mathbb{P}\{Q'=y\}}\right)$ is the privacy loss that is occurred when an adversary witnesses the outcome $y$.

---

[1] As shown in [Agrawal and Goyal, 2017], the Bernoulli reward setting can be generalized to any bounded reward setting.

## 2.3 RELATED WORK

In the classical stochastic bandit setting, the UCB-based, Thompson Sampling-based, and elimination-based algorithms all achieve good theoretical guarantees. Essentially, these algorithms rely on the empirical means to make decisions and the regret bounds take the $O(K \log(T)/\Delta)$ form.

As shown in Proposition 2.1 of [Dwork et al., 2014], differential privacy is invariant to post-processing, i.e., if a learning algorithm takes the output of an $\epsilon$-differentially private algorithm as input, then the output of this learning algorithm itself is also $\epsilon$-differentially private. In designing stochastic bandit algorithms with differential privacy, if the internal algorithm to compute the empirical mean is designed to be $\epsilon$-differentially private, then following from the post-processing property, we can claim the bandit algorithm itself is $\epsilon$-differentially private. This property has indeed been used in the design of private algorithms in previous work [Mishra and Thakurta, 2015, Tossou and Dimitrakakis, 2016, Sajed and Sheffet, 2019, Hu et al., 2021].

Mishra and Thakurta [2015] present the first differntially-private versions of the UCB and Thompson Sampling-based algorithms. However, the regret bounds they derive, $O\left(\frac{K \log^3(T)}{\epsilon \cdot \Delta}\right)$ and $O\left(\frac{K \log^3(T)}{\epsilon^2 \cdot \Delta^2}\right)$, are far from the $\Omega\left(\frac{K \log(T)}{\Delta} + \frac{K \log(T)}{\epsilon}\right)$ regret lower bound that is derived later by Shariff and Sheffet [2018]. The key reason for the sub-optimality is using the $T$-bounded Binary Mechanism[2] [Dwork et al., 2010, Chan et al., 2011] to add random noise to mask the empirical mean for an arm. Furthermore, since their algorithms need to know the time horizon $T$ in advance to calibrate the distribution of the random noise, they cannot be anytime learning algorithms. More importantly, their Thompson Sampling-based algorithm has some operational issues: in some rounds, the total reward $r_a(t)$ computed by the tree-based mechanism can take negative values, resulting in invalid parameters for the posterior distribution, the Beta distribution. Note that the parameters of Beta distributions must be non-negative. Our proposed learning algorithms carefully use clipping to address this issue.

Recently, two optimal algorithms have been proposed for differentially private stochastic bandits. Sajed and Sheffet [2019] propose DP-SE, an optimal elimination-style algorithm, and Hu et al. [2021] propose Anytime-Lazy-UCB, an optimal UCB-based algorithm. The key idea in achieving optimality is to use fresh observations to compute the differentially private empirical means, thus minimizing the number of noise variables needed. Although DP-SE, Anytime-Lazy-UCB, and our proposed Lazy-DP-TS are all optimal, as will be shown in Section 4, Lazy-DP-TS always outperforms the other two algorithms.

---

[2]Dwork et al. [2010], call it the Tree-based Mechanism, but the core idea is identical.

## 3 ALGORITHMS AND ANALYSIS

We now present our algorithms for achieving differential privacy in the stochastic bandit setting. The algorithms rely on two key ideas. The first is to use the differential privacy property of invariance to post-processing to make the arm selection algorithm differentially private due to the internal algorithm of computing empirical means being differentially private. The second is based on the principle of optimism in the face of uncertainty. Note that the decisions for the Thompson Sampling-based algorithms fully depend on the generated random samples from the posterior distributions. Operating under the optimism principle, we reshape the posterior distribution in an optimistic way: we shift the posterior distribution in the private algorithm towards the right as compared to the posterior distribution for the non-private Thompson Sampling. This shifting makes it more likely to draw a "good" posterior sample as compared to the draw in the non-private setting.

While both our algorithms rely on these fundamental concepts, the key difference between them lies in the design of the internal algorithm to compute the differentially private empirical means. Our first algorithm, Differentially Private Thompson Sampling (DP-TS), uses all the observations from the beginning in computing the differentially private empirical means, whereas our second algorithm, Lazy Differentially Private Thompson Sampling (Lazy-DP-TS) uses only a subsequence of observations.

Suppose at a given time step, arm $j$ has a sequence of $n$ observations $(x_1, x_2, \ldots, x_n)$. In DP-TS, all $n$ observations will be used to compute the differentially private empirical mean for arm $j$. We partition $(x_1, x_2, \ldots, x_n)$ into $(x_1, x_2, \ldots, x_m)$ and $(x_{m+1}, x_{m+1}, \ldots, x_n)$, where $m = 2^{\lfloor \log(n+1) \rfloor} - 1$. This partition guarantees $n - m \leq m$, i.e., the length of the first subsequence is always no smaller than the length of the second subsequence. The internal algorithm composes two differentially private mechanisms, each being $0.5\epsilon$-differentially private and acting on each partition, respectively, to process these $n$ observations.

The first mechanism is a modified version of the Logarithmic Mechanism [Chan et al., 2011] and works over $(x_1, x_2, \ldots, x_m)$: a differentially private aggregated reward of these $m$ observations will be computed. According to the original mechanism by Chan et al. [2011], random noise would be added to the reward of an arm whenever the number of observations of that arm hits $2^r$, for all $r \geq 0$, while in our modified version, random noise is added whenever the number of observations hits $2^{r+1} - 1$, so that fresh noise is added at longer epochs, resulting in less overall noise. The second mechanism is the bounded Binary Mechanism [Chan et al., 2011] and works over $(x_{m+1}, x_{m+2}, \ldots, x_n)$: random noise will be added based on the bounded Binary Mechanism and a differentially private aggregated reward of these $n - m$ observations will be output. The differentially

private empirical mean is thus computed by aggregating the outputs of these two mechanisms.

Note that a given observation may be used more than once in the calculation of the empirical means over rounds, which means more noise is required to maintain the same degree of privacy. Based on this remark, we propose Lazy-DP-TS, where the internal algorithm only uses a subsequence of all the observations obtained so far to compute the differentially private empirical mean and no observation can be reused, i.e., once an observation has been used, it will be abandoned. The length of the subsequences double each time, i.e., the internal algorithm adds a random noise to every $2^r, r \geq 0$ observations and outputs a differentially private empirical mean. This restriction of using an observation only once in the calculation of the empirical mean minimizes added noise and is thus the key to the optimality of differentially private online learning algorithms.

**Notation.** Let $\text{Beta}(\alpha, \beta)$ be a Beta distribution with parameters $\alpha, \beta$ and $\text{Lap}(b)$ be a Laplace distribution centered at 0 with scale $b$. The pdfs of $\text{Beta}(\alpha, \beta)$ and $\text{Lap}(b)$ are shown in Appendix. Also, $\log(x)$ is the base-2 logarithm of $x$ and $\ln(x)$ is the base-$e$ logarithm of $x$.

### 3.1 DP-TS

We now present DP-TS, followed by its guarantees.

#### 3.1.1 Algorithm

We first present some notation specific to this algorithm. $O_j(t-1) := \sum_{s=1}^{t-1} \mathbf{1}\{J_s = j\}$ counts the number of pulls of arm $j$ by the end of round $t-1$ and $\widehat{\mu}_{j,O_j(t-1)}$ is the empirical mean over these $O_j(t-1)$ observations. Let $\widetilde{\mu}_{j,O_j(t-1)}$ be the private empirical mean, i.e., $\widehat{\mu}_{j,O_j(t-1)}$ plus some noise.

DP-TS is presented in Algorithm 1. Lines 2 to 4 initialize the algorithm. We pull each arm once and set $\Psi_j = \{\}$ to hold future observations. Let $C_j$ track the differentially private aggregated reward computed by the modified Logarithmic Mechanism and $B_j$ track the private aggregated reward returned by the Binary Mechanism. Since for each arm the modified Logarithmic Mechanism processes observations in epochs, we use $r_j$ to index the arm-specific epoch, i.e., the modified Logarithmic Mechanism will add a noise variable to mask the aggregated reward of $2^{r_j}$ observations at the end of epoch $r_j$. We initialize $r_j = 0$ and the initialization phase adds random noise to the first observation.

Let $\upsilon_{\epsilon,O_j(t-1),t} := \frac{6\sqrt{8}\log(O_j(t-1)+1)\log(t)}{\epsilon \cdot O_j(t-1)}$. For all the rounds $t \geq K+1$, we first compute $\overline{\mu}_{j,O_j(t-1)} = \max\left\{0, \min\left\{\widetilde{\mu}_{j,O_j(t-1)} + \upsilon_{\epsilon,O_j(t-1),t}, 1\right\}\right\}$. Note that the empirical means are clipped so that $\overline{\mu}_{j,O_j(t-1)} \in [0,1]$.

---

**Algorithm 1** DP-TS

1: **Input:** Arm set $\mathcal{A}$ and privacy parameter $\epsilon$
2: **for** $t = 1, 2, \dots, K$ **do**
3:    Pull $J_t \leftarrow t$; Set $O_{J_t} \leftarrow 1$, $\Psi_{J_t} \leftarrow \{\}$, $C_{J_t} \leftarrow X_{J_t}(t) + \text{Lap}\left(\frac{1}{0.5\epsilon}\right)$, $r_{J_t} \leftarrow 0$, $B_{J_t} \leftarrow 0$, $\widetilde{\mu}_{J_t,O_{J_t}} \leftarrow \frac{C_{J_t}+B_{J_t}}{O_{J_t}}$
4: **end for**
5: **for** $t = K+1, K+2, \dots$ **do**
6:    **for** $j \in \mathcal{A}$ **do**
7:       Set $\overline{\mu}_{j,O_j}$
      $= \max\left\{0, \min\left\{\widetilde{\mu}_{j,O_j} + \frac{6\sqrt{8}\log(O_j+1)\log(t)}{\epsilon \cdot O_j}, 1\right\}\right\}$
8:       Set $\widetilde{\alpha}_j \leftarrow \overline{\mu}_{j,O_j} \cdot O_j$, $\widetilde{\beta}_j \leftarrow (1 - \overline{\mu}_{j,O_j}) \cdot O_j$
9:       Sample $\theta_j(t) \sim \text{Beta}(\widetilde{\alpha}_j + 1, \widetilde{\beta}_j + 1)$
10:    **end for**
11:    Pull arm $J_t \in \arg\max_{j \in \mathcal{A}} \theta_j(t)$
12:    Set $O_{J_t} \leftarrow O_{J_t} + 1$; Append $X_{J_t}(t)$ to $\Psi_{J_t}$
13:    **if** $O_{J_t} = \sum\limits_{s=0}^{r_{J_t}+1} 2^s$ **then**
14:       Set $C_{J_t} \leftarrow C_{J_t} + \sum \Psi_{J_t} + \text{Lap}\left(\frac{1}{0.5\epsilon}\right)$
15:       Set $\Psi_{J_t} \leftarrow \{\}$, $r_{J_t} \leftarrow r_{J_t} + 1$, $B_{J_t} \leftarrow 0$
16:    **else**
17:       Invoke $2^{r_{J_t}+1}$-bounded Binary Mechanism with Input $(0.5\epsilon, \Psi_{J_t})$ and Output $B_{J_t}$
18:    **end if**
19:    Set $\widetilde{\mu}_{J_t,O_{J_t}} \leftarrow \frac{C_{J_t}+B_{J_t}}{O_{J_t}}$ .
20: **end for**

---

We set $\widetilde{\alpha}_j(t) := \overline{\mu}_{j,O_j(t-1)} \cdot O_j(t-1)$ and $\widetilde{\beta}_j(t) := \left(1 - \overline{\mu}_{j,O_j(t-1)}\right) \cdot O_j(t-1)$. We then generate a random posterior sample $\theta_j(t) \sim \text{Beta}\left(\widetilde{\alpha}_j(t) + 1, \widetilde{\beta}_j(t) + 1\right)$ for each arm and pull the arm with the highest sample, i.e., $J_t \in \arg\max_{j \in \mathcal{A}} \theta_j(t)$. Since $\overline{\mu}_{j,O_j(t-1)} \in [0,1]$, the parameters of Beta distribution are valid.

To update the private empirical mean of the pulled arm, we append $X_{J_t}(t)$ to $\Psi_{J_t}$. If the number of observations in $\Psi_{J_t}$ hits $2^{r_{J_t}+1}$, we add random noise drawn from $\text{Lap}\left(\frac{1}{0.5\epsilon}\right)$ and update $C_{J_t}$. Since now all observations in $\Psi_{J_t}$ are used by the modified Logarithmic Mechanism, we reset $\Psi_{J_t}$ and $B_{J_t}$, and increment $r_{J_t}$ by one. If the number of observations in $\Psi_{J_t}$ has not reached $2^{r_{J_t}+1}$, we invoke the $2^{r_{J_t}+1}$-bounded Binary Mechanism [Chan et al., 2011] taking $\Psi_{J_t}$ as input and preserving $0.5\epsilon$-differential privacy. Note the number of observations in $\Psi_{J_t}$ is at most $2^{r_{J_t}+1}$.

**Remark.** (a) $r_j$ is determined by $O_j(t-1)$ as $r_j$ will only increment by one whenever the number of observations in $\Psi_j$ hits $2^{r_j+1}$. Indeed, $r_j = \lfloor\log(O_j(t-1)+1)\rfloor - 1$. (b) Regarding the noise variables included in the differentially private empirical mean, there are exactly $r_j+1$ i.i.d. random variables that are drawn from $\text{Lap}\left(\frac{1}{0.5\epsilon}\right)$ and at most $r_j+1$ i.i.d. random variables that are drawn from $\text{Lap}\left(\frac{r_j+1}{0.5\epsilon}\right)$.

We now compare Algorithm 1 to the non-private Thompson Sampling by Agrawal and Goyal [2017]. Let $\alpha'_j(t) := \widehat{\mu}_{j,O_j(t-1)} \cdot O_j(t-1)$ be the number of successes and $\beta'_j(t) := \left(1 - \widehat{\mu}_{j,O_j(t-1)}\right) \cdot O_j(t-1)$ be the number of failures among $O_j(t-1)$ Bernoulli trials. Recall that in the non-private Thompson Sampling, we draw $\theta'_j(t) \sim \text{Beta}\left(\alpha'_j(t)+1, \beta'_j(t)+1\right)$. By adding $v_{\epsilon,O_j(t-1),t}$ to $\widetilde{\mu}_{j,O_j(t-1)}$, we have, with high probability, $\overline{\mu}_{j,O_j(t-1)} \geq \widehat{\mu}_{j,O_j(t-1)}$, i.e., the posterior distribution for the differentially private version is shifted towards the right as compared to the non-private version.

### 3.1.2 Analysis

We present privacy and regret guarantees for Algorithm 1.

**Theorem 1.** *Algorithm 1 is $\epsilon$-differentially private.*

*Proof.* We first show the internal algorithm to compute the empirical mean, i.e., from Lines 12 to 19, is $\epsilon$-differentially private. Then, from Proposition 2.1 of Dwork et al. [2014], we conclude that Algorithm 1 is $\epsilon$-differentially private. Note that Lines 6 to 11 can be viewed as post-processing since in these steps, the learning algorithm does not touch any revealed observations. Suppose reward sequences $\boldsymbol{X}$ and $\boldsymbol{X}'$ differ in round $h$, i.e., the reward vectors $X_h = (X_1(h), \ldots, X_K(h))$ and $X'_h = (X'_1(h), \ldots, X'_K(h))$ are not the same. Note that changing from $X_h$ to $X'_h$ has no impact on other arms except arm $J_h$ as only the reward of the pulled arm, $J_h$, is revealed in round $h$. Let $J_h = j$. At the end of round $h$, the differentially private empirical mean of arm $j$ will be updated. According to Algorithm 1, changing from $X_j(h)$ to $X'_j(h)$ impacts $C_j$ by at most 1. From Theorem 3.6 of Dwork et al. [2014], we know the internal algorithm to compute $C_j$ (Line 14) is $0.5\epsilon$-differentially private. From Theorem 3.5 of Chan et al. [2011], we know the internal algorithm to compute $B_j$ (Line 17) is $0.5\epsilon$-differentially private. Composing these two internal algorithms together, from Theorem 3.14 in [Dwork et al., 2014], we conclude that the internal algorithm (Line 19) to compute the differentially private empirical mean is $\epsilon$-differentially private. □

**Theorem 2.** *The regret $\mathcal{R}_{DP\text{-}TS}(T)$ of Algorithm 1 is at most*

$$\sum_{j \in \mathcal{A}:\Delta_j > 0} O\left(\max\left\{\frac{\log(T)}{\Delta_j}, \frac{\log(T)}{\epsilon}\log\left(\frac{\log(T)}{\epsilon \cdot \Delta_j}\right)\right\}\right) \quad .$$

**Remark.** Several remarks are in order. (a): DP-TS is optimal up to a $\log\log(T)$ factor. (b): When setting $\epsilon \to \infty$, Algorithm 1 boils down to the same algorithm as the one by Agrawal and Goyal [2017]. However, our derived regret bound, Theorem 2, is only order-optimal instead of asymptotically optimal. Note that the regret bound of the non-private Thompson Sampling can be asymptotically optimal, i.e., a regret bound attaining the best possible coefficient for the leading term asymptotically. (c): Algorithm 1 also has an

$O\left(\sqrt{KT\log(T)} + \frac{K\log(T)}{\epsilon}\log\left(\frac{\sqrt{T\log(T)}}{\sqrt{K}\epsilon}\right)\right)$ problem-independent regret bound. Note that it is known that Thompson Sampling is able to achieve the $\Omega\left(\sqrt{KT}\right)$ minimax lower bound for non-private stochastic bandits [Jin et al., 2021]. Therefore, the $O\left(\sqrt{KT\log(T)}\right)$ term in Theorem 2 is $\sqrt{\log(T)}$ far from being minimax optimal. Note that the price of introducing differential privacy is $\Omega\left(\frac{K\log(T)}{\epsilon}\right)$ [Shariff and Sheffet, 2018]. This lower bound implies DP-TS is $\log\left(\frac{\sqrt{T\log(T)}}{\sqrt{K}\epsilon}\right)$ far from being optimal in the private setting. The detailed proof for the problem-independent result is deferred to Appendix.

We now provide a proof sketch for Theorem 2. The detailed proof is deferred to Appendix. Let $\mathcal{F}_{t-1}$ collect all the history information containing the pulled arms, the rewards associated with the pulled arms, and the added noise. Define $\mathcal{F}_0 = \{\}$. Let $y_j := \mu_1 - \frac{\Delta_j}{6}$ and define $E_j^\theta(t)$ as the event that $\{\theta_j(t) \leq y_j\}$. Let $C_j(t-1)$ be the event that $\left\{|\mu_j - \widehat{\mu}_{j,O_j(t-1)}| \leq \sqrt{\frac{3\log(t)}{O_j(t)}}\right\}$. Let $G_j(t-1)$ be the event that $\left\{|\widehat{\mu}_{j,O_j(t-1)} - \widetilde{\mu}_{j,O_j(t-1)}| \leq v_{\epsilon,O_j(t-1),t}\right\}$.

*Proof sketch of Theorem 2.* We upper bound $\mathbb{E}[O_j(T)]$. Let $\mathcal{L}_j := \max\left\{\frac{108\log(T)}{\Delta_j^2}, \frac{72\log(T)}{\epsilon \cdot \Delta_j}\log\left(\frac{72\log(T)}{\epsilon \cdot \Delta_j}\right)\right\}$. We separate all $T$ rounds into two regimes based on whether $O_j(t-1) \geq \mathcal{L}_j$. For all rounds $t$ s.t. $O_j(t-1) < \mathcal{L}_j$, the total regret is at most $\mathcal{L}_j \cdot \Delta_j$. In a round when $O_j(t-1) \geq \mathcal{L}_j$, w.t.p., we have $\overline{\mu}_{j,O_j(t-1)} \leq \mu_j + \frac{4\Delta_j}{6}$, which implies $\overline{E_j^\theta(t)}$ is a low probability event. Meanwhile, w.h.p., we also have $\overline{\mu}_{1,O_1(t-1)} \geq \widehat{\mu}_{1,O_1(t-1)}$, which allows us to reduce the proof to the non-private setting.

With these ideas in hand, we have $\sum_{t=1}^T \mathbb{E}\left[\mathbf{1}\{J_t = j\}\right]$

$$\leq \mathcal{L}_j + \underbrace{\sum_{t=1}^T \mathbb{P}\left\{\overline{C_j(t-1)}\right\} + \sum_{t=1}^T \mathbb{P}\left\{\overline{G_j(t-1)}\right\}}_{=:\omega_0}$$

$$+ \underbrace{\sum_{t=1}^T \mathbb{P}\left\{O_j(t-1) > \mathcal{L}_j, C_j(t-1), G_j(t-1), \overline{E_j^\theta(t)}\right\}}_{=:\omega_1}$$

$$+ \underbrace{\sum_{t=1}^T \mathbb{P}\left\{J_t = j, E_j^\theta(t)\right\}}_{=:\omega_2} \quad .$$

(1)

Via well-known concentration inequalities, we have $\omega_0 \leq O(1)$ (lemmas are shown in Appendix). For $\omega_1$, we use the argument that if events $C_j(t-1)$ and $G_j(t-1)$ are true simultaneously and arm $j$ has been pulled at least $\mathcal{L}_j$ times, we have $\overline{\mu}_{j,O_j(t-1)} \leq \mu_j + \frac{4\Delta_j}{6}$. Since $\theta_j(t) \sim \text{Beta}\left(\widetilde{\alpha}_j(t)+1, \widetilde{\beta}_j(t)+1\right)$, from the properties

of the Beta distribution, we know that it is very unlikely to draw $\theta_j(t) > \mu_j + \frac{5\Delta_j}{6}$. In Appendix, we show that $\omega_1 \leq O(1)$.

The key challenge is to upper bound $\omega_2$. We first reduce the proof to the non-private Thompson Sampling. Then, we reuse Lemmas 2.9 and 2.10 in [Agrawal and Goyal, 2017] to conclude the proof. Now, we show how to reduce the proof to the non-private setting. By introducing $G_1(t-1)$ and $\overline{G_1(t-1)}$, term $\omega_2$ is at most

$$\sum_{t=1}^{T} \mathbb{P}\left\{J_t = j, G_1(t-1), E_j^\theta(t)\right\} + \sum_{t=1}^{T} \mathbb{P}\left\{\overline{G_1(t-1)}\right\}.$$

For the second term above, it is at most $O(1)$ (shown in Appendix). For the first term above, we have

$$\sum_{t=1}^{T} \mathbb{P}\left\{J_t = j, G_1(t-1), E_j^\theta(t)\right\}$$
$$\leq \mathbb{E}\left[\sum_{t=1}^{T} \frac{\mathbb{P}\{\theta_1(t) \leq y_j | \mathcal{F}_{t-1}\}}{1 - \mathbb{P}\{\theta_1(t) \leq y_j | \mathcal{F}_{t-1}\}} \{J_t = 1, G_1(t-1)\}\right]$$
$$\leq \mathbb{E}\left[\sum_{t=1}^{T} \frac{\mathbb{P}\{\theta_1'(t) \leq y_j | \mathcal{F}_{t-1}\}}{1 - \mathbb{P}\{\theta_1'(t) \leq y_j | \mathcal{F}_{t-1}\}} \{J_t = 1\}\right],$$
(2)

where $\theta_1'(t) \sim \text{Beta}\left(\alpha_j'(t) + 1, \beta_j'(t) + 1\right)$, the non-private posterior distribution for arm $j$ conditioned on $\mathcal{F}_{t-1}$.

The first inequality in (2) links the probability of pulling a sub-optimal $j$ to the probability of pulling the best arm by using a lemma that we develop in Appendix. The last inequality uses the fact that if $\overline{\mu}_{1,O_1(t-1)} \geq \widehat{\mu}_{1,O_1(t-1)}$, we have $\mathbb{P}\{\theta_1(t) \leq y_j \mid \mathcal{F}_{t-1}\} \leq \mathbb{P}\{\theta_1'(t) \leq y_j \mid \mathcal{F}_{t-1}\}$, i.e., $\text{Beta}\left(\widetilde{\alpha}_j(t) + 1, \widetilde{\beta}_j(t) + 1\right)$ stochastically dominates $\text{Beta}\left(\alpha_j'(t) + 1, \beta_j'(t) + 1\right)$. Since the proof now is reduced to the non-private setting, slightly modifying Lemmas 2.9 and 2.10 in [Agrawal and Goyal, 2017] concludes the proof. In Appendix, we show $\omega_2 \leq O\left(\frac{\log(T)}{\Delta_j^2}\right)$. □

## 3.2 LAZY-DP-TS

We now present Lazy-DP-TS and its guarantees. The idea to achieve optimality is limiting the number of times an observation is used in computing the empirical mean to one.

### 3.2.1 Algorithm

We first present some notation specific to this algorithm. Let $O_j(t-1)$ denote the number of observations that are used to compute the differentially private empirical mean and $\widehat{\mu}_{j,O_j(t-1)}$ denote the empirical mean of these $O_j(t-1)$ observations. Let $\widetilde{\mu}_{j,O_j(t-1)}$ be the differentially private empirical mean.

Lazy-DP-TS is presented in Algorithm 2. Lines 2 to 4 are the initialization. We pull each arm once and add random noise that is drawn from $\text{Lap}\left(\frac{1}{\epsilon}\right)$ to the obtained observation

---

**Algorithm 2** Lazy-DP-TS

1: **Input:** Arm set $\mathcal{A}$ and privacy parameter $\epsilon$
2: **for** $t = 1, 2, \ldots, K$ **do**
3:     Pull $J_t \leftarrow t$; Set $O_{J_t} \leftarrow 1, \widetilde{\mu}_{J_t,O_{J_t}} \leftarrow X_{J_t}(t) + \text{Lap}\left(\frac{1}{\epsilon}\right), r_{J_t} \leftarrow 0, \Psi_{J_t} \leftarrow \{\}$
4: **end for**
5: **for** $t = K+1, K+2, \ldots$ **do**
6:     **for** $j \in \mathcal{A}$ **do**
7:         Set $\overline{\mu}_{j,O_j} = \max\left\{0, \min\left\{\widetilde{\mu}_{j,O_j} + \frac{3\log(t)}{\epsilon \cdot O_j}, 1\right\}\right\}$
8:         Set $\widetilde{\alpha}_j \leftarrow \overline{\mu}_{j,O_j} \cdot O_j, \widetilde{\beta}_j \leftarrow (1 - \overline{\mu}_{j,O_j}) \cdot O_j$
9:         Sample $\theta_j(t) \sim \text{Beta}(\widetilde{\alpha}_j + 1, \widetilde{\beta}_j + 1)$
10:     **end for**
11:     Pull $J_t \in \arg\max_{j \in \mathcal{A}} \theta_j(t)$
12:     Append $X_{J_t}(t)$ to $\Psi_{J_t}$
13:     **if** number of observations in $\Psi_{J_t}$ hits $2^{r_{J_t}+1}$ **then**
14:         Set $O_{J_t} \leftarrow 2^{r_{J_t}+1}, \widetilde{\mu}_{J_t,O_{J_t}} \leftarrow \frac{\sum \Psi_{J_t} + \text{Lap}\left(\frac{1}{\epsilon}\right)}{O_{J_t}}$
15:         Set $r_{J_t} \leftarrow r_{J_t} + 1, \Psi_{J_t} \leftarrow \{\}$
16:     **end if**
17: **end for**

---

to initialize the differentially private empirical mean. We still use $2^{r_j}$ to track the number of observations that have been used to compute the differentially private empirical mean for arm $j$. Initially, we set $r_j = 0$ and $\Psi_j = \{\}$ to hold future observations.

For all rounds $t \geq K+1$, we first compute $\overline{\mu}_{j,O_j(t-1)} = \max\left\{0, \min\left\{\widetilde{\mu}_{j,O_j(t-1)} + \frac{3\log(t)}{\epsilon \cdot O_j(t-1)}, 1\right\}\right\}$ and then compute $\widetilde{\alpha}_j(t) := \overline{\mu}_{j,O_j(t-1)} \cdot O_j(t-1)$ and $\widetilde{\beta}_j(t) := \left(1 - \overline{\mu}_{j,O_j(t-1)}\right) \cdot O_j(t-1)$. Next, we generate a posterior sample $\theta_j(t) \sim \text{Beta}\left(\widetilde{\alpha}_j(t) + 1, \widetilde{\beta}_j(t) + 1\right)$ for each arm and pull the arm with the highest posterior sample, i.e., $J_t \in \arg\max_{j \in \mathcal{A}} \theta_j(t)$.

To process $X_{J_t}(t)$, we append it in $\Psi_{J_t}$. However, we may not update the differentially private empirical mean of the pulled arm in round $t$. We will only update it when the number of observations in $\Psi_{J_t}$ hits $2^{r_{J_t}+1}$ and the updated differentially private empirical mean will be based on observations in $\Psi_{J_t}$ only, i.e., the updated differentially private empirical mean is computed by adding a noise variable drawn from $\text{Lap}\left(\frac{1}{\epsilon}\right)$ to these fresh $2^{r_{J_t}+1}$ observations. Since now all observations in $\Psi_{J_t}$ are used, we reset $\Psi_{J_t}$ and increment $r_{J_t}$ by one.

**Remark.** (a) The number of observations used to compute the differentially private empirical mean doubles each time, i.e., $O_j(t-1)$ takes values from $2^{r_j}, r_j \geq 0$. (b) The number of noise variables included in the private empirical mean of arm $j$ is always 1 and it is drawn from $\text{Lap}\left(\frac{1}{\epsilon}\right)$.

### 3.2.2 Analysis

We now present privacy and regret guarantees for Algorithm 2.

**Theorem 3.** *Algorithm 2 is $\epsilon$-differentially private.*

*Proof.* The internal algorithm to compute the differentially private empirical mean is shown in Lines 12 to 16 in Algorithm 2. Lines 5 to 11 can be viewed as post-processing. Now, we show that the internal algorithm is $\epsilon$-differentially private. Suppose reward sequences $\boldsymbol{X}$ and $\boldsymbol{X}'$ differ in round $h$, i.e., the reward vectors $X_h = (X_1(h), \ldots, X_K(h))$ and $X_h' = (X_1'(h), \ldots, X_K'(h))$ are not the same. The changing from $X_h$ to $X_h'$ can only impact arm $J_h$. Let $J_h = j$. Since arm $j$'s differentially private means are always based on fresh observations, the changing from $X_h$ to $X_h'$ can only impact the differentially private aggregated reward of arm $j$ once and by at most 1. By adding a noise variable drawn from $\mathrm{Lap}\left(\frac{1}{\epsilon}\right)$ to $\sum \Psi_j$, from Theorem 3.6 in [Dwork et al., 2014], we know that the internal algorithm to compute the differentially private empirical mean is $\epsilon$-differentially private. $\square$

**Theorem 4.** *The regret $\mathcal{R}_{\text{Lazy-DP-TS}}(T)$ of Algorithm 2 is at most* $\sum_{j \in \mathcal{A}: \Delta_j > 0} O\left(\frac{\log(T)}{\min\{\epsilon, \Delta_j\}}\right)$.

**Remark**. Several remarks are in order. (a): Lazy-DP-TS is (order)-optimal as its regret upper bound matches the
$$\Omega\left(\sum_{j \in \mathcal{A}: \Delta_j > 0} \frac{\log(T)}{\Delta_j} + \frac{\log(T)}{\epsilon}\right)$$
regret lower bound of Shariff and Sheffet [2018]. Our Lazy-DP-TS preserves the same regret guarantee as the one for Anytime-Lazy-UCB by Hu et al. [2021] and DP-SE by Sajed and Sheffet [2019]. However, as will be shown in Section 4, Lazy-DP-TS has better practical performance than Anytime-Lazy-UCB and DP-SE. Since Algorithm 2 drops observations as it learns, even if we set $\epsilon \to \infty$, the regret bound can never be asymptotically optimal. (b): Algorithm 2 also has an $O\left(\sqrt{KT\log(T)} + \frac{K\log(T)}{\epsilon}\right)$ problem-independent regret bound. Since the price of introducing differential privacy is $\Omega\left(\frac{K\log(T)}{\epsilon}\right)$, the $O\left(\frac{K\log(T)}{\epsilon}\right)$ term in Theorem 4 cannot be improved as it matches the lower bound of introducing differential privacy. Therefore, Lazy-DP-TS is minimax optimal up to a $\sqrt{\log(T)}$ factor in both private setting and non-private setting. The detailed proof for the problem-independent result is deferred to Appendix.

We now present a proof sketch for Theorem 4. The detailed proof is deferred to Appendix. We still define $C_j(t-1)$ as the event that the confidence interval of the empirical mean holds and $G_j(t-1)$ as the event that the noise injected is not too much. Let $\mathcal{F}_{t-1}$ collect all the history information and set $y_j := \mu_1 - \frac{\Delta_j}{6}$. Let event $E_j^\theta(t) := \{\theta_j(t) \le y_j\}$.

*Proof sketch of Theorem 4.* We still upper bound the expected number of pulls of a sub-optimal arm $j$. However, we cannot separate all $T$ rounds into two regimes since Algorithm 2 drops observations. Instead, we perform a decomposition as follows.

$$
\begin{aligned}
&\sum_{t=1}^{T} \mathbb{E}\left[\mathbf{1}\left\{J_t = j\right\}\right] \\
\le\ & \underbrace{\sum_{t=1}^{T} \mathbb{P}\left\{J_t = j, C_j(t-1), G_j(t-1), \overline{E_j^\theta(t)}\right\}}_{=:\omega_1} \\
&+ \underbrace{\sum_{t=1}^{T} \mathbb{P}\left\{J_t = j, E_j^\theta(t), G_1(t-1)\right\} + O(1)}_{=:\omega_2}.
\end{aligned}
\tag{3}
$$

Note that the $O(1)$ term in (3) is an upper bound on $\sum_{t=1}^{T} \mathbb{P}\left\{\overline{C_j(t-1)}\right\} + \mathbb{P}\left\{\overline{G_j(t-1)}\right\} + \mathbb{P}\left\{\overline{G_1(t-1)}\right\}$.

To upper bound $\omega_1$, we let $\mathcal{L}_j := \frac{72 \cdot \log(T)}{\Delta_j \cdot \min\{\epsilon, \Delta_j\}}$ and $d_j := \log(\mathcal{L}_j)$. Recall that for arm $j$, the numbers of observations that are used to compute the differentially private empirical means are $2^{r_j}$ for $0 \le r_j \le \log(T)$. Let $\tau_{r_j}$ be the round such that at the end of round $\tau_{r_j}$, the learning algorithm will use $2^{r_j}$ observations to update the differentially private empirical mean for arm $j$. We separate $0 \le r_j \le \log(T)$ into two parts. The first part is when $0 \le r_j \le d_j$. Based on the definition of $\tau_{r_j}$, we know that the total number of pulls of arm $j$ is at most $\sum_{s=0}^{d_j} 2^s \le O\left(\frac{\log(T)}{\Delta_j \cdot \min\{\epsilon, \Delta_j\}}\right)$ in all rounds up to (and including) $\tau_{d_j}$. When $d_j < r_j \le \log(T)$, we have $2^{r_j} > \mathcal{L}_j$, i.e., we have accumulated "enough" observations for arm $j$. For a fixed $r_j$, with high probability, the expected number of pulls of arm $j$ is at most $O(1)$ in all rounds $t \in \{\tau_{r_j} + 1, \ldots, \tau_{r_j+1}\}$. Then, we know that the total expected number of pulls is at most $O(\log(T))$ in all rounds from $\tau_{d_j} + 1$ up to $T$. In Appendix, we show $\omega_1 \le O\left(\frac{\log(T)}{\Delta_j \cdot \min\{\epsilon, \Delta_j\}}\right)$.

The challenge still lies in upper bounding $\omega_2$. We again use the ideas shown in (2) to reduce the proof to the non-private setting. We have

$$
\omega_2 \le \mathbb{E}\left[\sum_{t=1}^{T} \frac{\mathbb{P}\{\theta_1'(t) \le y_j | \mathcal{F}_{t-1}\}}{1 - \mathbb{P}\{\theta_1'(t) \le y_j | \mathcal{F}_{t-1}\}} \{J_t = 1\}\right].
\tag{4}
$$

However, now we cannot reuse Lemmas 2.9 and 2.10 from [Agrawal and Goyal, 2017] directly due to the fact that the observations for arm 1 are also dropped during the learning. To tackle this challenge, we separate all $T$ rounds into multiple intervals based on whether arm 1's empirical mean is updated or not. Let $\tau_r$ be the round such that at the end of round $\tau_r$, the learning algorithm will use $2^r$ observations for arm 1 to update arm 1's empirical mean, i.e., in all rounds $t \in \{\tau_r + 1, \ldots, \tau_{r+1}\}$, the posterior distribution

for $\theta'_1(t)$ stays the same. Then, we have

$$
\begin{aligned}
\omega_2 &\leq \mathbb{E}\left[\sum_{r=0}^{\log(T)}\sum_{t=\tau_r+1}^{\tau_{r+1}}\frac{\mathbb{P}\{\theta'_1(t)\leq y_j|\mathcal{F}_{t-1}\}}{\mathbb{P}\{\theta'_1(t)>y_j|\mathcal{F}_{t-1}\}}\{J_t=1\}\right] \\
&= \sum_{r=0}^{\log(T)}\mathbb{E}\left[\frac{\mathbb{P}\{\theta'_1(\tau_r+1)\leq y_j|\mathcal{F}_{\tau_r}\}}{\mathbb{P}\{\theta'_1(\tau_r+1)>y_j|\mathcal{F}_{\tau_r}\}}\sum_{t=\tau_r+1}^{\tau_{r+1}}\{J_t=1\}\right] \\
&\leq \sum_{r=0}^{\log(T)}2^{r+1}\cdot\mathbb{E}\left[\frac{\mathbb{P}\{\theta'_1(\tau_r+1)\leq y_j|\mathcal{F}_{\tau_r}\}}{\mathbb{P}\{\theta'_1(\tau_r+1)>y_j|\mathcal{F}_{\tau_r}\}}\right].
\end{aligned}
\tag{5}
$$

The last inequality uses the fact that the number of pulls for arm 1 in all rounds $t \in \{\tau_r + 1, \ldots, \tau_{r+1}\}$ is at most $2^{r+1}$ based on the definition of $\tau_{r+1}$. Let $d_1 := \log\left(\frac{8}{\mu_1-y_j}\right)$. We now analyze two cases separately based on whether $0 \leq r \leq \lfloor d_1 \rfloor$ or $r \geq \lceil d_1 \rceil$. By using Lemma 2.9 of Agrawal and Goyal [2017] and other analysis, we have $\sum_{r=0}^{\lfloor d_1 \rfloor}2^{r+1}\mathbb{E}\left[\frac{\mathbb{P}\{\theta'_1(\tau_r+1)\leq y_j|\mathcal{F}_{\tau_r}\}}{\mathbb{P}\{\theta'_1(\tau_r+1)>y_j|\mathcal{F}_{\tau_r}\}}\right] \leq O\left(\frac{1}{\Delta_j^2}\right)$ and $\sum_{r=\lceil d_1 \rceil}^{\log(T)}2^{r+1}\mathbb{E}\left[\frac{\mathbb{P}\{\theta'_1(\tau_r+1)\leq y_j|\mathcal{F}_{\tau_r}\}}{\mathbb{P}\{\theta'_1(\tau_r+1)>y_j|\mathcal{F}_{\tau_r}\}}\right] \leq O\left(\frac{\log(T)}{\Delta_j^2}\right)$. In Appendix, we show $\omega_2 \leq O\left(\frac{\log(T)}{\Delta_j^2}\right)$. $\square$

# 4 EXPERIMENTAL RESULTS

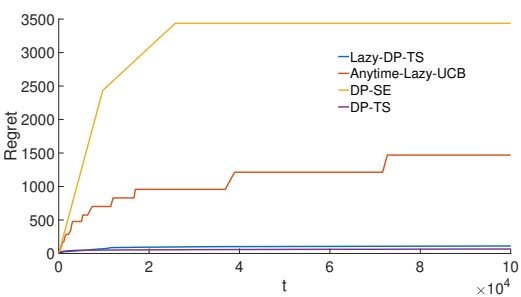

Figure 1: $\epsilon = 500$

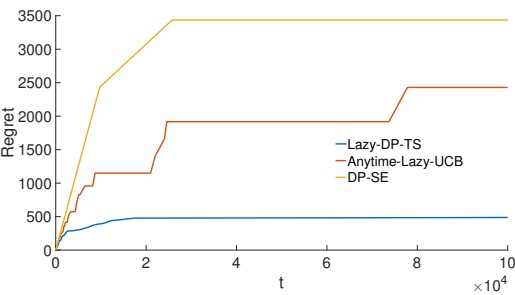

Figure 2: $\epsilon = 1.0$

We compare the practical performance among DP-TS, Lazy-DP-TS, DP-SE, and Anytime-Lazy-UCB under the experimental setting that has been used in [Sajed and Sheffet, 2019], i.e., we have $K = 5$ arms with mean rewards setting to $0.75, 0.625, 0.5, 0.375, 0.25$ and the privacy parameter $\epsilon$

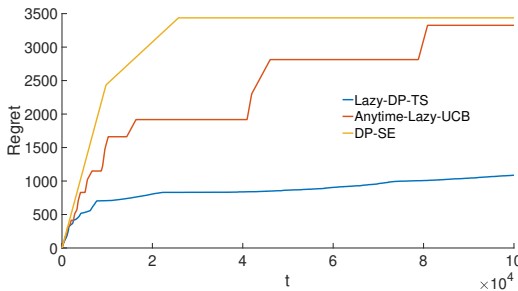

Figure 3: $\epsilon = 0.5$

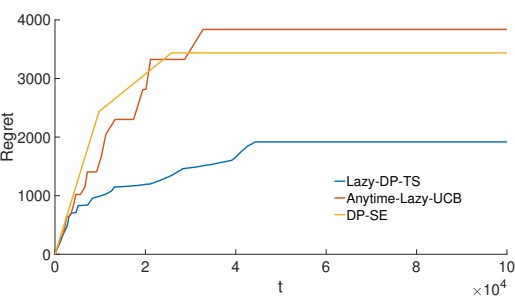

Figure 4: $\epsilon = 0.25$

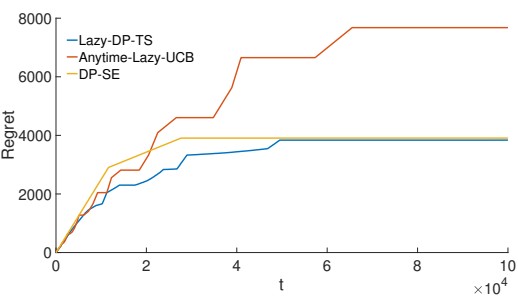

Figure 5: $\epsilon = 0.10$

setting to $0.1, 0.25, 0.5, 1.0, 500$. We set $T = 10^5$. Figure 1 shows the results of the setting where $\epsilon = 500$. It is not surprising that DP-TS outperforms Lazy-DP-TS as when $\epsilon$ is very large, DP-TS is asymptotically optimal while Lazy-DP-TS can only be order-optimal. Also, just as expected, Thompson Sampling-based algorithms outperform the UCB-based and elimination-style algorithms. Figures 2 to 5 show the results of the settings where $\epsilon = 1.0, 0.5, 0.25, 0.1$, we skip the plots of DP-TS as the practical performance of DP-TS is inferior to the remaining three optimal algorithms when $\epsilon$ is very small. From the experimental results we can see that Lazy-DP-TS always outperforms DP-SE and Anytime-Lazy-UCB. More experimental results, including comparison of private and non-private algorithms, can be found in Appendix.

# 5  CONCLUSION

We have presented optimal Thompson Sampling-based algorithms for differentially private stochastic bandits, filling a gap in the literature for differentially private online learning. The ideas used in this paper also contribute to developing optimal algorithms for other settings such as differentially private combinatorial multi-armed bandits [Chen et al., 2020]. Note that both the UCB and elimination-based algorithms are deterministic. So far, our proposed algorithms have not used the unique feature that only Thompson Sampling-based algorithms have, the randomness inherent in the learning algorithms. An interesting future direction is the design of optimal private Thompson Sampling-based algorithms using the fact that a random posterior sample may provide a degree of differential privacy for free [Wang et al., 2015, Foulds et al., 2016].

## ACKNOWLEDGEMENTS

This work is supported by Amii Post-Doctoral Fellowships.

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
