# OpenReview forum: "Near-Optimal Thompson Sampling-based Algorithms for Differentially Private Stochastic Bandits"
_auai.org/UAI/2022/Conference — UAI 2022 Poster_

### Official Review · Reviewer_eNXv · 2022-04-11

**Q2(1) Originality/Novelty:** 3
**Q2(2) Significance/Impact:** 2
**Q2(3) Correctness/Technical Quality:** 3
**Q2(6) Clarity Of Writing:** 3
**Q6 Overall Score:** 6
**Q8 Confidence In Your Score:** 3

**Q1 Summary And Contributions:**

This paper considers the differential privacy stochastic MAB setting. Their setting crucially depends on Definition 1, which states that for any two neighboring reward vectors that only differ in one component the internal algorithm will output almost similar decisions. They propose two algorithms DP-TS and Lazy-DP-TS that uses this differential privacy and reaches order optimal regret. This is the first provable TS algorithm for this setting. They evaluate their algorithms in some toy settings.

**Q2 Assessment Of The Paper:**

More detailed information regarding each of these aspects is given below:

**Q2(4) Quality Of Experiments (Optional):**

2: Fair: The experimental evaluation is weak: important baselines are missing, or the results do not adequately support the main claims.

**Q2(5) Reproducibility:**

3: Good: Key resources (e.g., proofs, code, data) are available and key details (e.g., proofs, experimental setup) are sufficiently well-described for competent researchers to confidently reproduce the main results.

**Q3 Main Strengths:**

1) The differential privacy setting is an important area and they propose the first order optimal Thompson Sampling algorithm for this setting.

2) The exposition to their algorithm is clear and they also theoretically analyze their algorithm. The Lazy-DP-TS is also order optimal and reaches the lower bound proposed in Shariff and Sheffet [2018].

3) They empirically analyze their algorithm on toy datasets.

**Q4 Main Weakness:**

1) It seems to be that the regret analysis mostly follows the same track as Agrawal and Goyal [2017], with some minor changes. What is the key technical improvement/novelty over their proof?

2) Similarly, Theorem 2 and Theorem 4 showing DP-TS and Lazy-DP-TS are differentially private seems to be the extension of the proofs from Dwork et al., 2014. What is the key technical improvement/novelty over their proof?

3) Writing can be improved. You need to discuss the implication of Definition 1. Especially why there is $e^{\epsilon}$, and not just $\epsilon$. It is also not clear to me why you choose Laplacian distribution to set the value of $C_j$ the cumulative differential privacy reward tracker. Can you explain it?

**Q5 Detailed Comments To The Authors:**

Refer to 1), 2), 3) in Q4.

4) The role of $v_{\epsilon, O_{j}(t-1), t}$ is not clear to me. Is it like an optimistic estimation of the mean, that shifts the posterior to the right?

5) Then again, it seems that you clip the optimistic estimation of the mean $\bar{\mu}_{j}$. I don't think that Agrawal and Goyal [2017] use any clipped mean estimation. How do you handle this in your proof sketch? Is that the key improvement over the proof technique of Agrawal and Goyal [2017]?

6) The toy experiments really don't bring out much. It would be nice to have some experiments on a real dataset, like Movielens/Yahoo which are standard datasets used in the MAB setting.

**Q7 Justification For Your Score:**

Refer to Q3, Q4, Q5. If the authors answer my queries sufficiently in Q4, Q5, I am willing to raise my scores.

**Q9 Complying With Reviewing Instructions:**

1: Yes.

---

### Official Review · Reviewer_bfNz · 2022-04-13

**Q2(1) Originality/Novelty:** 2
**Q2(2) Significance/Impact:** 2
**Q2(3) Correctness/Technical Quality:** 3
**Q2(6) Clarity Of Writing:** 4
**Q6 Overall Score:** 6
**Q8 Confidence In Your Score:** 4

**Q1 Summary And Contributions:**

See below.

**Q2 Assessment Of The Paper:**

More detailed information regarding each of these aspects is given below:

**Q2(4) Quality Of Experiments (Optional):**

3: Good: The experimental evaluation is adequate, and the results convincingly support the main claims.

**Q2(5) Reproducibility:**

2: Fair: Key resources (e.g., proofs, code, data) are unavailable but key details (e.g., proof sketches, experimental setup) are sufficiently well-described for an expert to confidently reproduce the main results.

**Q3 Main Strengths:**

The work proposes the first TS-type algorithm for differentially private stochastic bandits.
Due to the better practical performance of TS, such an attempt is worthwhile.
The regret upper bound of the proposed DP-lazy-TS algorithm matches the problem lower bound.  Some experiments are conducted to show the advantage of TS-based algorithms over UCB and elimination-type ones.

**Q4 Main Weakness:**

The main weakness is the technical contribution. Given the previous analysis of Anytime-Lazy-UCB (Hu et al, 2021), the main difficulty is to analyze the regret caused by inaccurate samples of the optimal arm 1. This paper deal with this difficulty by simply reshaping the posterior distribution optimistically and then using the standard non-private TS analysis on this part. Based on the above consideration, the technical contribution is a bit weak.

**Q5 Detailed Comments To The Authors:**

Q1:

The paper studies the differentially private stochastic bandit problem and proposes TS-based algorithms to solve the problem. The main idea to obtain the regret analysis while guaranteeing the DP is to reshape the posterior distribution in an optimistic way such that the mean of the posterior can be better than that in the non-private setting. They propose both DP-TS and Lazy-DP-TS, the latter matches the regret lower bound. Some experiments are conducted to verify the efficiency of the TS-based algorithm compared with UCB and elimination-based ones.

Q5:

The current technique to deal with TS is reshaping the posterior distribution and thus previous analysis for non-private TS can be used. I would like to see more discussions in the paper on the difficulty of TS itself for the DP problem.

Some minor issues:

The regret with expectation over actions and rewards should be called expected regret?
In experiments, how the reward is generated and the number of runs are not reported.



**Q7 Justification For Your Score:**

I mainly consider the strength of the work. Trying to give a positive result of TS for the DP problem is worthwhile. Though the technique of reshaping the posterior distribution is simple, it does not bring additional cost on complexity or other aspects.

**Q9 Complying With Reviewing Instructions:**

1: Yes.

---

### Official Review · Reviewer_zjEC · 2022-04-13

**Q2(1) Originality/Novelty:** 2
**Q2(2) Significance/Impact:** 2
**Q2(3) Correctness/Technical Quality:** 3
**Q2(6) Clarity Of Writing:** 3
**Q6 Overall Score:** 6
**Q8 Confidence In Your Score:** 4

**Q1 Summary And Contributions:**

The paper proposes an algorithm for stochastic bandit learning in a private manner. The proposed algorithms are DP-TS and Lazy-DP-TS. DP-TS uses a composition of Logarithmic Mechanism and bounded Binary Mechanism while Lazy-DP-TS uses laplacian noise addition.
Regret analysis is provided for both the algorithms, where DP-TS is shown
to be optimal till an O(log log(T)) factor. Performance improvement Lazy-DP-TS is illustrated through experiments.

**Q2 Assessment Of The Paper:**

More detailed information regarding each of these aspects is given below:

**Q2(5) Reproducibility:**

3: Good: Key resources (e.g., proofs, code, data) are available and key details (e.g., proofs, experimental setup) are sufficiently well-described for competent researchers to confidently reproduce the main results.

**Q3 Main Strengths:**

• Useful regret bounds are provided. Especially for DP-TS it is shown that
the regret is optimal till an O(log log(T)) factor.
• It is identified that using the same reward observation for multiple estimations will require more noise for the same degree of privacy

**Q4 Main Weakness:**

In theorem 2 and 4, equivalence between differential privacy guarantees
for computing empirical mean in 1 round and differential privacy of overall
the algorithm is used. A composition overall differentially private empirical
mean calculation needs to be shown to state the algorithm as differentially
private (the privacy factor also changes in this case).
• A more technical analysis needs to be shown to demonstrate that using
same reward observation multiple times will require more noise.

**Q5 Detailed Comments To The Authors:**

Theorems should be numbered starting from 1, not 2.
• Composition over all empirical mean estimation needs to be shown to state
DP −T S as differentially private since the same reward observations would
be used repeatedly. Also, the complete algorithm will have a privacy factor
exceeding ϵ because of composition.

**Q7 Justification For Your Score:**

The algorithms for Thompson sampling and differentially privacy are not novel, only their combined usage in a manner so as to
be optimal in terms of regret is novel.
Significance/Impact: The algorithm suggested are setting specific and it is
non-trivial to generalize the algorithms for other settings.
Correctness/Technical Quality: Regret proofs for both algorithms are technically thorough.
Reproducibility: The algorithms are presented in an easily implementable mannner

**Q9 Complying With Reviewing Instructions:**

1: Yes.

---

### Decision · Program_Chairs · 2022-05-15

**Decision:**

Accept (Poster)

**Comment:**

Meta Review: This paper proposes nearly optimal Thompson sampling-based algorithms with differentially privacy guarantees. The authors have provided a very detailed and helpful response to the reviewer and meta reviewer's questions. There is unanimous support to accept this paper. Thus, I recommend acceptance.